# Association of microRNA-210-3p with NT-proBNP, sST2, and Galectin-3 in heart failure patients with preserved and reduced ejection fraction: A cross-sectional study

Jasmine Chandra Arul[1], Sudagar Singh Raja Beem[2], Mohanalakshmi Parthasarathy[3], Mahesh Kumar Kuppusamy[4], Karthikeyan Rajamani[5], Santhi Silambanan[1]*

**1** Department of Biochemistry, Sri Ramachandra Institute of Higher Education and Research, Chennai, Tamil Nadu, India, **2** Department General Medicine, Sri Ramachandra Institute of Higher Education and Research, Chennai, Tamil Nadu, India, **3** Department of Biochemistry, Sri Muthukumaran Medical College Hospital and Research Institute, Chennai, Tamil Nadu, India, **4** Department of Physiology and Biochemistry, Government Yoga and Naturopathy Medical College and Hospital, Chennai, Tamil Nadu, India, **5** Department of Public Health, Sri Ramachandra Institute of Higher Education and Research, Chennai, Tamil Nadu, India

* santhisilambanan@sriramachandra.edu.in, santhisilambanan@gmail.com

## Abstract

### Background

Heart failure (HF) is a growing health problem and around two percent are affected in the general population. Accurate diagnostic markers that have the potential for early diagnosis of HF are lacking. This study aimed to compare the expression levels of microRNA-210-3p with biomarkers NT-proBNP, sST2, and galectin-3, in heart failure patients with preserved and reduced ejection fractions.

### Materials and methods

The cross-sectional study was conducted on 270 hypertensive heart failure patients in the age group of 30 to 75 years of both genders. The participants with evidence of HF were recruited from the Department of Cardiology in a tertiary care hospital in Chennai, India. MicroRNA-210-3p was analyzed by qRT-PCR in a stratified sample of 80 HF patients and 20 apparently healthy individuals. Biomarkers were analyzed by ELISA. Institutional ethics committee approval and written informed consent were obtained. Statistical analysis was performed using R software (4.2.1). Based on the type of distribution of data, appropriate statistical tools were used. p-value ≤ 0.05 was considered to be statistically significant.

### Results

All the biomarkers including microRNA-210-3p were significantly higher in HFrEF than in HFpEF. MAGGIC score showed a positive correlation with all the biomarkers. The cut-off of microRNA-210-3p was 5.03.

**Data availability statement:** Relevant data are located at Dryad: https://doi.org/10.5061/dryad.qjq2bvqs4.

**Funding:** By JCA U02B180535 From Sri Ramachandra University Funding sources help in data collection and analysis.

**Competing interests:** The authors have declared that no competing interests exist.

**Abbreviations:** ACE, angiotensin-converting enzyme; AHA/ACC, American Heart Association/ American College of Cardiology; AMI, acute myocardial infarction; AUC, area under the curve; BMI, body mass index; BNP, brain natriuretic peptide; CKD, chronic kidney disease; COACH study, Coordinating Study Evaluating Outcomes of Advising and Counseling in Heart Failure; CSI-KHFR, Cardiology Society of India-Kerala Acute Heart Failure Registry; CT, cycle threshold; CV, coefficient of variation; CVD, cardiovascular disease; ECG, electrocardiogram; ECHO, echocardiography; EF, ejection fraction; ELISA, enzyme-linked immunosorbent assay; ESC, European Society of Cardiology; FC, fold change; Gal-3, Galectin-3; GDF-15, Growth Differentiation Factor-15; GFR, glomerular filtration rate; HbA1c, glycated hemoglobin A1c; HDL, high-density lipoprotein; HF, heart failure; HFmrEF, heart failure with mid-range or mildly reduced ejection fraction; HFpEF, heart failure with preserved ejection fraction; HFrEF, heart failure with reduced ejection fraction; HIF-1α, hypoxia-inducible factor- 1 alpha; IL-33, interleukin-33; JNC, Joint National Committee; LDL, low-density lipoprotein; LV, left ventricle; LVH, left ventricle hypertrophy; MAGGIC, Meta-Analysis Global Group in Chronic Heart Failure; MAPK, mitogen-activated protein kinase; miRNA, microRNA; NF-κB, nuclear factor – kappa beta; NP, natriuretic peptides; NT-proBNP, N-terminal pro-brain natriuretic peptide; NYHA, New York Heart Association; qRT-PCR, quantitative reverse transcription-polymerase chain reaction; RAAS, renin–angiotensin–aldosterone system; RNA, ribonucleic acid; ROC, receiver operating characteristics; RV, right ventricle; SCD, sudden cardiac death1; sST2, soluble suppression of tumorigenicity 2; T2DM, type 2 diabetes mellitus; TGL, triglyceride; WHR, waist-hip ratio; YLD, years lost due to disability.

## Conclusion

All the biomarkers were significantly elevated in HFrEF compared to HFpEF. However, microRNA-210-3p could be an early marker in the diagnosis of heart failure. The strategy of employing a multi-marker approach could help in the early diagnosis as well as in stratifying the HF patients.

## Introduction

Heart failure (HF) is a complex multisystem disorder due to structural and functional alterations in the heart [1]. It affects 64.34 million people, leading to 9.91 million years lost due to disability (YLDs). In the US, HF prevalence has been projected to increase from 2.42% in 2012 to 2.97% in 2030 [2]. The prevalence of HF in Asia is 1.3 - 6.7%, which is higher than that of Western countries [3]. Among the Asian countries, the prevalences are 1.3%, 1.0%, 6.7%, and 4.5% in China, Japan, Malaysia, and Singapore respectively [4]. In India, the number of new HF cases may increase from 0.118 - 0.708 million in 2000 to 0.214 -1.3 million in 2025 [5].

HF patients are grouped into three phenotypes based on the ejection fraction (EF) of the left ventricle (LV), which is measured by echocardiography (ECHO). They are HF with reduced EF (HFrEF) with EF ≤ 40%, HF with preserved EF (HFpEF) with EF ≥ 50%, and HF with mid-range/ mildly reduced EF (EFmrEF) with EF 41-49% [6]. HFrEF is preceded by acute myocardial infarction (AMI), valvular, and other cardiac diseases [7]. Hypertension is the most important cause of HFpEF, with a prevalence of 60 - 89% among HF patients [7]. HFpEF is prevalent among the older population, especially women. With the advent of new diagnostic and treatment modalities, the prevalence of HFrEF is declining, while that of HFpEF shows an increasing trend [8].

The diagnosis of HF is based on the Framingham Diagnostic Criteria for HF [9]. Diagnosis and management of HFrEF have been studied in detail and have been implemented effectively into practice [8]. However, HFpEF and HFmrEF are poorly investigated and managed, particularly in the developing countries [10]. The European Society of Cardiology (ESC) guidelines (2021) recommend that the patients suspected to have acute HF should have their plasma NPs tested. However, the chances of missing the early-stage HF could be high, especially in situations where EF is more than 50% as in HFpEF [7]. American Heart Association (AHA)/ American College of Cardiology (ACC) guidelines recommend the measurement of soluble suppression of tumorigenicity 2 (sST2) for risk stratification in patients with acute HF [11]. Galectin-3 (Gal-3) is involved in inflammation, repair, and fibrosis in HF [12].

Despite the introduction of various markers in the diagnosis and prognosis of HF, there is still a gap in the management due to heterogeneity in the presentation of HF. Recent studies have shown that circulating microRNA (miR) is playing a crucial role in the pathogenesis and progression of HF, thus, having a great potential to be diagnostic as well as prognostic markers. According to Rincón et al. there is an association between the microRNAs such as miR-210-3p, miR-221-3p, and miR-23a-3p and cardiac morbidity and mortality [13]. In cardiac tissue, miR-210-3p is upregulated in tissue hypoxic conditions; thus, regulating cell differentiation, proliferation, migration, apoptosis, mitochondrial metabolism, and angiogenesis [14]. Studies on the utility of miRNAs in the diagnosis and prognosis of HFpEF are limited. This study was aimed to determine the association of miR-210-3p, and biomarkers such as NT-proBNP, sST2, and Gal-3 in heart failure patients with preserved and reduced ejection fractions.

## Materials and methods

The prospective cross-sectional study was conducted among 270 HF patients of both genders in the age group of 30 to 75 years. The sample size was calculated based on the study by Spinar et al. 2019 [15]. Hypertensive patients with evidence of HF were recruited from the Department of Cardiology at Sri Ramachandra Institute of Higher Education and Research, Chennai, India from 01-12-2020 to 15-02-2023. Initially, the study was decided to be conducted only on hypertensive patients with HF. Due to the COVID-19 pandemic, recruiting patients with hypertension alone was highly challenging. Hence, with the advice of the research advisory committee members, it was decided to include hypertensive HF patients who also had diabetes mellitus; and revised ethics approval accordingly. Thus, among the 270 patients, 168 patients had diabetes mellitus in addition to hypertension. Diagnosis of HF was based on Framingham Heart Failure Diagnostic criteria [9].

The study participants were subjected to transthoracic 2D Doppler echocardiography (Phillips and GE Healthcare echocardiography) with patients lying supine in the left lateral decubitus position. According to the definition of European and United States guidelines, the normal EF is 52-72% in men and 54-74% in women [16]. According to Chengode, EF > 55% is considered to be normal. It can be 45–54%, 30 - 44%, and < 30% in mild, moderate, and severe LV dysfunction respectively [17]. Anthropometric characteristic such as body mass index (BMI) in $Kg/m^2$ was measured as per the Asia-Pacific guidelines [18]. Systolic blood pressure (SBP) and diastolic blood pressure (DBP) were measured using standard procedure and patients were identified as hypertensives based on the JNC7 blood pressure classification [19]. According to the International Expert Committee, diabetes mellitus was diagnosed based on glycated hemoglobin (HbA1c) ≥6.5%, fasting plasma glucose ≥ 126 mg/dL, and 2-hour plasma glucose or random plasma glucose ≥ 200 mg/dL [20].

In the present study, EF of 50% was used as the cut-off to categorize the participants into two groups group 1 – HF with EF ≤ 49% and group 2 -HF with EF ≥ 50%. The participants were grouped into group 1 with HF patients with EF ≤ 49% and group 2 with EF ≥50%. Patients with acute HF in the past 3 months, AMI in the past 6 weeks, disorders of thyroid, lung, renal, and liver, pregnancy, cancer, systemic infectious diseases, connective tissue disorders, and patients on anticancer drugs, steroids, and oral contraceptive pills were excluded from the study. The Meta-Analysis Global Group in Chronic Heart Failure (MAGGIC) mortality risk score was calculated in all the study participants. The MAGGIC score consists of thirteen clinical predictor variables such as age, sex, BMI, SBP, EF, serum creatinine level, participant was a current smoker, diabetes mellitus, and chronic obstructive pulmonary disease, belongs to New York Heart Association (NYHA) class, HF diagnosed for at least eighteen months, use of beta-blockers and angiotensin-converting enzyme (ACE) inhibitors.

NT-pro BNP, sST2, and Gal-3 were measured by ELISA:

- NT-proBNP: measurement range: 7.5 - 120 pg/mL, sensitivity: 0.5 pg/mL, CV%: intra-assay < 9% and inter-assay < 11%; Biological Reference Interval: ≤74 years: < 125 pg/mL, and ≥ 75 years: < 450 pg/mL [16].

- sST2: measurement range: 0.31-20.0 ng/mL, sensitivity: 0.19 ng/mL, CV%: intra-assay 5.4% and inter-assay 6.1%, Biological Reference Interval: 14.42 ng/mL [17].

- Gal-3: measurement range: 0.16-10.0 ng/mL, sensitivity: 0.16ng/mL, CV%: intra-assay 5.56% and inter-assay 7.27%, Biological Reference Interval: < 17.8 ng/mL [18].

MiR (MicroRNA)-210-3p was analyzed by qRT-PCR. MiR-210-3p was expressed as cycle threshold folds. Analysis of microRNA-210-3p: Methodology: RT-PCR, microRNA was isolated by PureFast® microRNA mini spin purification kit [19].

> hsa‑miR‑210‑3p MIMAT0000267; CUGUGCGUGUGACAGCGGCUGA

Analysis was done on 100 samples; with 20 controls (apparently healthy individuals without hypertension or diabetes mellitus or HF), 40 patients of group 1 and 40 patients of group 2. Fold change was calculated as follows:

$$\Delta CT = CT - CT$$

$$\Delta\Delta CT = \Delta CT - \Delta CT_{control}$$

$$FC = 2^{-\Delta\Delta CT}$$ (CT : cycle threshold, FC : fold change)

## Ethics statement

Ethics approval was obtained from the institutional ethics committee. Ethics number: IEC-NI/19/FEB/68/09, dated 10.11.2020. Written informed consent was obtained from all the study participants.

## Statistics used

Statistical analysis was performed using R software version 4.2.1. The obtained data were subjected to Kolmogorov-Smirnov test to check for normality of distribution. Continuous variables were expressed as mean and standard deviation or median and interquartile range and the student 't' test or Mann-Whitney U test was used to compare the data between the groups. Categorical variables were expressed as frequency and percentage; the Chi-Square test or Fischer-exact test was used to compare the data. The Pearson correlation coefficient was used to test the correlation among the variables. The receiver operating characteristics (ROC) curve was used to find the area under the curve (AUC). The Youden index was used to find the cut-off values of biomarkers. P value $\leq 0.05$ was considered statistically significant.

Large Language Models (LLMs), such as ChatGPT were not used by any of the authors in this manuscript.

## Results

Based on the EF cut-off of 50%, 170 patients have EF $\geq$ 50%, while the remaining 100 HF patients have reduced EF of $\leq$ 49%. Fig 1 shows the flow chart indicating the recruitment of participants into the study along with patient characteristics.

Majority of patients were males; in group 1, they were older, 61-75 years, whereas in group 2, the majority were middle-aged. In both the groups, most of them were in obese category. According to NYHA class, participants in group 1 were in NYHA classes III and IV, whereas in group 2, the participants were in classes I and II (p < 0.001) (Table 1).

MicroRNA, including the biomarkers and the MAGGIC score, were higher in HFrEF than in HFpEF. HFrEF had higher total cholesterol levels than HFpEF (p = 0.01). (Table 2).

To analyze the influence of gender, age, BMI, and diabetes mellitus in heart failure, the groups are further classified. Tables 3–5 show the distribution of patients among the two

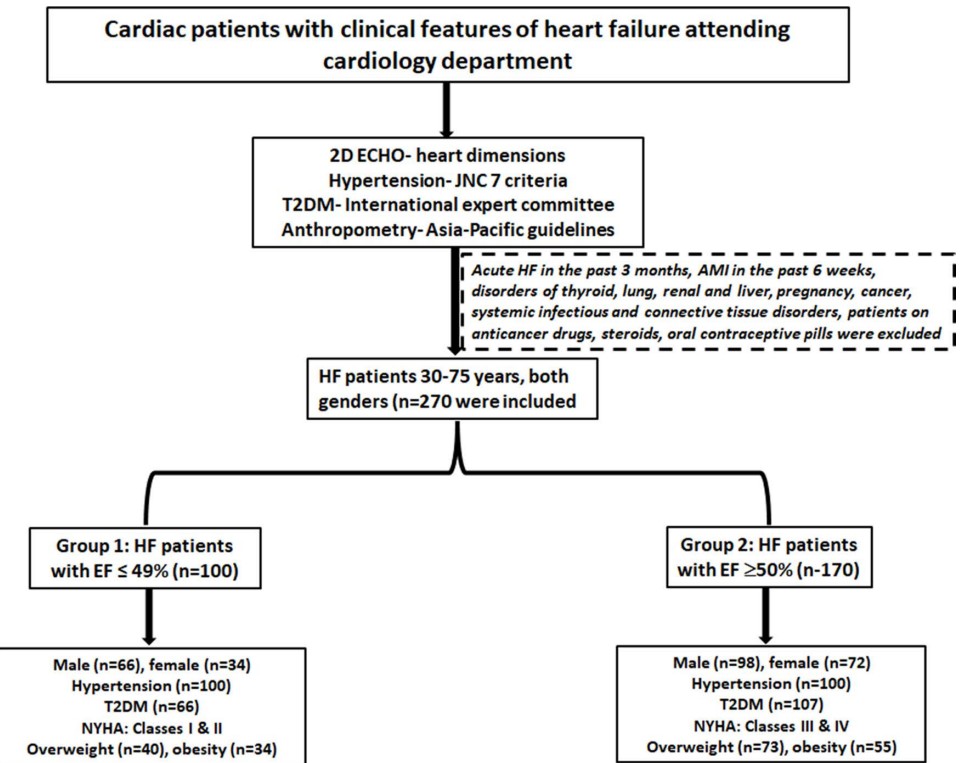

**Fig 1. Shows the flow chart displaying the recruitment of study participants and their characteristics.**

groups based on gender, age (cut-off of 50 years), and presence/absence of diabetes mellitus respectively.

The cardiac biomarkers did not show statistically significant alterations based on gender in both the groups. Females showed statistically decreased MAGGIC score in HFrEF (p = 0.01) and HFpEF (p = 0.003). Females with HFpEF had statistically significant alteration in tri-glyceride level (p = 0.002), whereas individuals in HFrEF did not show significant alteration in lipid profile in HFrEF. (Table 3).

In both the groups, the distribution of patients more than 51 years was prevalent; however, statistical significance was obtained in group 2 (p = 0.03). Among the cardiac biomarkers, only NT-proBNP showed statistically significant difference in HFpEF (p = 0.02) in both the groups. The MAGGIC score showed statistically significant difference in both the groups (P < 0.001) (Table 4).

The mean age of the participants with diabetes was higher than nondiabetics in both HFrEF (p = 0.039) and HFpEF (p = 0.040). Number of individuals with diabetes was higher than nondiabetics and most of them were males (p = 0.003) and (p = 0.004) in HFrEF and HFpEF respectively. The glycemic variables such as fasting plasma glucose (p < 0.001), post-prandial plasma glucose (p < 0.001), and glycated hemoglobin (p < 0.001) among diabetics in both the groups. All the cardiac biomarkers including MiR-210-3p did not show statistically significant difference between diabetics and non-diabetics in HFrEF and HFpEF. The MAG-GIC showed higher values in diabetics compared to nondiabetics in HFrEF (p = 0.014) and HFpEF (p < 0.001) (Table 5).

The analysis of biomarkers for heart failure prediction revealed that the MAGGIC score demonstrated the best overall performance with the highest AUC of 0.86 (p = 0.001), a cut-off

**Table 1. The demographic data of the study participants.**

| Characteristics | Group 1 (n = 100) HF with EF ≤ 49% | Group 2 (n = 170) HF with EF ≥ 50% | p-value |
|---|---|---|---|
| Age (Years) | | | |
| < 45 n (%) | 12 (12) | 5 (2) | <0.001** |
| 46-60 n (%) | 26 (26) | 118 (70) | |
| 61 -75 n (%) | 62 (62) | 47 (28) | |
| Gender | | | |
| Female n (%) | 34 (34) | 72 (42) | 0.21 |
| Male n (%) | 66 (66) | 98 (58) | |
| Common presenting clinical features | Breathlessness, pedal edema, and palpitation | Breathlessness, fatigue | – |
| BMI (Kg/m²) | | | |
| Normal weight n (%) | 22 (22) | 33 (19) | 0.39 |
| Overweight n (%) | 13 (13) | 33 (19) | |
| Obese n (%) | 65 (65) | 104 (62) | |
| T2DM | | | |
| Present n (%) | 66 (66) | 102 (60) | 0.37 |
| Absent n (%) | 34 (34) | 68 (40) | |
| NYHA Class (n/%) | III (59/59) | I (160/94) | <0.001** |
| | IV (41/41) | II (10/6) | |

BMI: body mass index; T2DM: type 2 diabetes mellitus; NYHA: New York Heart Association; Expressed as frequency & percentage; Chi-Square test was done;

**: p-value statistically high significant.

**Table 2. The levels of biomarkers among the participants.**

| Characteristics | Group 1 (n = 100) HF with EF ≤ 49% | Group 2 (n = 170) HF with EF ≥ 50% | p-value |
|---|---|---|---|
| TC (mg/dL) | 205.29 (46.42) | 193.72 (38.19) | 0.01* |
| TG (mg/dL) | 155.80 (63.06) | 149.55 (73.58) | 0.41 |
| HDL (mg/dL) | 43.74 (13.00) | 42.76 (10.38) | 0.92 |
| LDL (mg/dL) | 126.02 (39.78) | 124.28 (33.26) | 0.54 |
| NT-proBNP (pg/mL) | 482.74 (110.94) | 287.8 (81.62) | 0.001** |
| sST2 (ng/mL) | 38.24 (13.15) | 21.40 (7.42) | 0.001** |
| Gal-3 (ng/mL) | 24.93 (10.84) | 17.77 (6.03) | 0.001** |
| MiR-210-3p (folds) | 11.89 (31.11) | 1.08 (1.36) | 0.03* |
| MAGGIC score (points) | 22.94 (5.92) | 8.05 (4.64) | 0.001* |

TC: total cholesterol; TG: triglyceride; HDL: high-density lipoprotein, LDL: low-density lipoprotein, NT-proBNP: N-terminal- pro-brain natriuretic peptide, sST2: soluble suppression of tumorigenicity 2y, Gal-3: Galectin-3; MiR: micro ribonucleic acid; MAGGIC: Meta-Analysis Global Group in Chronic Heart Failure.

Expressed as mean and SD; Student t-test was used;

**: p-value statistically high significant

*: p-value statistically significant.

**Table 3. The levels of biomarkers among the participants based on gender.**

| | Group 1: HF with EF ≤ 49% (n = 100) | | p-value | Group 2: HF with EF ≥ 50% (n = 170) | | p-value |
|---|---|---|---|---|---|---|
| | Male(n = 66) | Female (n = 34) | | Male (n = 98) | Female(n = 72) | |
| Age | 62.09 (11.25) | 61.18 (12.02) | 0.70 | 53.63 (10.91) | 55.69 (9.86) | 0.21 |
| NYHA class (n/%) @ | III (36/22) | III (23/22) | 0.20 | I (92/56) | I (68/64) | 0.87 |
| | IV (30/18) | IV (11/10) | | II (6/4) | II (4/4) | |
| TC | 208.53(45.87) | 199.0 (47.52) | 0.33 | 196.39 (39.67) | 190.10(36.04) | 0.29 |
| TG | 152.74(63.20) | 161.74 (63.30) | 0.50 | 164.45 (85.23) | 129.26(47.30) | 0.002** |
| HDL | 44.21(13.15) | 42.82 (12.85) | 0.61 | 41.90 (10.14) | 43.93 (10.65) | 0.21 |
| LDL | 129.83 (38.69) | 118.62 (41.40) | 0.18 | 125.81 (34.90) | 122.21(31.01) | 0.48 |
| FPG | 133.65(64.89) | 146.50 (63.98) | 0.34 | 130.95 (49.67) | 121.44(37.91) | 0.17 |
| PPPG | 200.57(92.42) | 223.58 (91.58) | 0.23 | 169.44 (86.10) | 160.26(72.38) | 0.46 |
| HbA1c | 7.83 (2.75) | 8.43 (2.36) | 0.28 | 6.81 (1.70) | 6.62 (1.38) | 0.43 |
| NT-proBNP | 483.09(66.35) | 482.05(168.08) | 0.96 | 284.92 (92.59) | 291.75(95.11) | 0.63 |
| sST2 | 37.97 (12.01) | 38.77 (15.30) | 0.77 | 19.98 (9.41) | 23.33 (18.19) | 0.12 |
| Gal-3 | 25.40 (10.69) | 24.0 (11.22) | 0.54 | 16.94 (7.86) | 18.91 (8.71) | 0.12 |
| MiR-210-3p# | 1.32 (0.14-5.51) | 2.16 (0.54-34.43) | 0.38 | 0.28 (0.15-0.73) | 0.14 (0.09-0.95) | 0.18 |
| MAGGIC Score | 24.0 (5.65) | 20.88 (5.96) | 0.01* | 8.93 (4.67) | 6.85(4.34) | 0.003** |

NYHA: New York Heart Association; TC: total cholesterol; TG: triglyceride; HDL: high-density lipoprotein, LDL: low-density lipoprotein; FPG: fasting plasma glucose; PPPG: post-prandial plasma glucose, HbA1c: Glycated hemoglobin; NT-proBNP: N-terminal- pro-brain natriuretic peptide, sST2: soluble suppression of tumorigenicity 2; Gal-3: Galectin-3; MiR: micro ribonucleic acid; MAGGIC: Meta-Analysis Global Group in Chronic Heart Failure.

Expressed as mean and SD; Student t-test was used; #: expressed as median and interquartile range; Mann-Whitney U test was used; @: Chi-square test was used;

**: p-value statistically high significant

*: p-value statistically significant.

score of 14 with sensitivity of 95% and specificity of 89%. MiR-210-3p showed AUC of 0.79, sensitivity of 87%, and specificity of 54% at a cut-off level of 5.03 (Table 6, Fig 4).

MAGGIC score showed a positive correlation with NT-proBNP (p < 0.001), sST2 (p < 0.001), and galectin-3 (p < 0.001). NT-proBNP (p < 0.001), sST2 (p < 0.012), and MAGGIC (p < 0.001) scores positively correlated with HbA1c (Table 7).

## Discussion

Heart failure (HF) is the consequence of various cardiac and noncardiac conditions [1]. After the sixth decade of life, the incidence of HF doubles in men whereas it triples in women [21]. The Cardiology Society of India-Kerala Acute Heart Failure Registry (CSI-KHFR) indicates that India is facing a huge threat of HF due to its large population, and increased prevalence of diabetes mellitus, obesity, hypertension, pulmonary diseases, and sedentary lifestyle [22]. The states such as Kerala, Punjab, and Tamil Nadu are the most affected [23]. In the present study, in group 1 (HF ≤ 49%), 62% of patients were in the advancing age of 61 - 75 years, while in group 2 (HF ≥ 50%), 70% were in 46 - 60 years. In group 2, the earlier onset of HF is due to the presence of underlying comorbid conditions. Males were predominantly affected irrespective of EF. Generally, males are at risk for the development of comorbid conditions such as obesity, hypertension and T2DM. In this study, most of the participants in both the groups were obese. This could be the cause or edema as a consequence of HF. According to NYHA class, participants in group 1 were in NYHA classes III and IV, whereas in group 2, the participants were in classes I and II (p < 0.001) (Table 1). The mean age of the participants with diabetes was higher than nondiabetics in both HFrEF (p = 0.039) and HFpEF (p = 0.040). Number of individuals

**Table 4. The levels of biomarkers among the participants of ≤ 50 and ≥ 51years.**

| | Group 1: HF with EF ≤ 49% (n = 100) | | p-value | Group 2: HF with EF ≥ 50% (n = 170) | | p-value |
|---|---|---|---|---|---|---|
| | Age ≤ 50 (n = 17) | Age ≥ 51yrs (n = 83) | | Age ≤ 50 yrs (n = 61) | Age ≥ 51yrs (n = 109) | |
| Male (n/%) | 10 (10) | 56 (56) | 0.49 | 42 (27) | 57 (33) | **0.03*** |
| Female (n/%) | 7 (7) | 27 (27) | | 19 (10) | 52 (30) | |
| NYHA class (n/%) @ | III (13/17) | III (46/24) | 0.10 | I (58/74) | I (103/54) | 0.86 |
| | IV (4/5) | IV (37/19) | | II (3/4) | II (6/3) | |
| TC | 208.33 (39.65) | 204.55 (47.87) | 0.76 | 201.23 (38.70) | 189.52 (37.43) | 0.054 |
| TG | 160.59 (59.16) | 154.82 (64.12) | 0.73 | 163.02 (77.77) | 142.01 (70.37) | 0.07 |
| HDL | 44.94 (11.91) | 43.49 (13.26) | 0.67 | 43.13 (13.37) | 42.55 (8.31) | 0.72 |
| LDL | 128.88 (30.63) | 125.43 (41.54) | 0.74 | 130.62 (31.99) | 120.73 (33.57) | 0.06 |
| FPG | 141.86 (81.47) | 136.95 (58.75) | 0.77 | 128.0 (48.32) | 126.62 (43.87) | 0.84 |
| PPPG | 269.88 (30.63) | 200.53 (79.50) | **<0.001**** | 163.50 (78.22) | 166.10 (81.22) | 0.83 |
| HbA1c | 7.49 (2.98) | 8.14 (2.55) | 0.35 | 6.62 (1.68) | 6.79 (1.51) | 0.50 |
| NT-proBNP | 426.66 (50.15) | 494.22 (116.65) | **0.02*** | 266.33 (94.24) | 299.83 (91.24) | **0.02*** |
| sST2 | 43.02 (17.12) | 37.26 (12.07) | 0.10 | 20.97 (10.09) | 21.64 (15.64) | 0.76 |
| Gal-3 | 27.39 (10.53) | 24.42 (10.90) | 0.30 | 17.62 (8.86) | 17.86 (7.99) | 0.85 |
| MiR-210-3p# | 0.53 (0.05-3.01) | 1.89 (0.16-14.26) | 0.76 | 0.23 (0.13-2.38) | 0.25 (0.10-0.87) | 0.88 |
| MAGGIC Score | 16.65 (4.61) | 24.23 (5.32) | **<0.001**** | 5.52 (2.16) | 9.46 (5.03) | **<0.001**** |

NYHA: New York Heart Association; TC: total cholesterol; TG: triglyceride; HDL: high-density lipoprotein, LDL: low-density lipoprotein; FPG: fasting plasma glucose; PPPG: post-prandial plasma glucose, HbA1c: Glycated hemoglobin; NT-proBNP: N-terminal- pro-brain natriuretic peptide, sST2: soluble suppression of tumorigenicity 2; Gal-3: Galectin-3; MiR: micro ribonucleic acid; MAGGIC: Meta-Analysis Global Group in Chronic Heart Failure;

Expressed as mean and SD; Student t-test was used; #: expressed as median and interquartile range; Mann-Whitney U test was used; @: Chi-square test was used;

**: p-value statistically high significant

*: p-value statistically significant.

with diabetes was higher than nondiabetics and most of them were males (p = 0.003) and (p = 0.004) in HFrEF and HFpEF respectively (Table 5). According to Teramoto et al, the mean age of HF patients in India is 58 - 68 years. The prevalence of HFpEF is higher than that of HFrEF. HFpEF is higher in women, while HFrEF is higher in men. HFpEF phenotype tends to increase with the aging of the population, especially of more than 64 years [24].

The MAGGIC score is used to predict mortality as well as stratify the patients with HF. It can be done with the routinely available parameters. However, the MAGGIC score is not able to predict the frequency of hospitalization, and morbidity [25]. In the present study, the MAGGIC score showed statistically significant difference (p = 0.001) when both the groups were compared (Table 2). Females showed statistically decreased MAGGIC score in HFrEF (p = 0.01) and HFpEF (p = 0.003) (Table 3). The score showed statistically significant difference in both the groups when compared according to age of above and below 50 years in both the groups (P < 0.001) (Table 4). The score showed higher values in diabetics compared to nondiabetics in HFrEF (p = 0.014) and HFpEF (p < 0.001) (Table 5). Thus, the MAGGIC score had shown the influence of gender, age and presence of diabetes among patients with hypertensive HF.

In the present study, when the variables were compared among HFrEF and HFpEF, miR-210-3p (p = 0.03), NT-proBNP (p = 0.001), sST2 (p = 0.001), and Gal-3 (p = 0.001) were higher in HFrEF than HFpEF. HFrEF had higher total cholesterol levels than HFpEF (p = 0.01) (Table 2). All the patients were hypertensives; moreover, 66% in HFrEF and 40% in HFpEF had diabetes also (Table 1). Females with HFpEF had statistically significant lower triglyceride levels (p = 0.002), whereas individuals in HFrEF did not show significant alteration in lipid profile (Table 3). Hypertension, diabetes and obesity alter the endothelial function

**Table 5. The Levels of biomarkers among the groups with and without DM.**

| Characteristics | Group 1: HF with EF ≤ 49% (n = 100) | | p-value | Group 2: HF with EF ≥ 50% (n = 170) | | p-value |
|---|---|---|---|---|---|---|
| | DM present (n = 66) | DM absent (n = 34) | | DM present (n = 102) | DM absent (n = 68) | |
| Age | 63.47(10.97) | 58.50(11.84) | **0.039***| 56.46(9.34) | 53.21(11.06) | **0.040*** |
| Gender@ | | | | | | |
| Male (n/%) | 37 (37) | 29 (29) | **0.003*** | 39 (23) | 59 (35) | **0.004*** |
| Female (n/%) | 29 (29) | 05 (05) | | 29 (17) | 43 (25) | |
| NYHA class (n/%)@ | III-37 | III-22 | 0.40 | I-65 | I-95 | 0.50 |
| | IV-29 | IV-12 | | II- 3 | II-7 | |
| TC | 203.08(50.00) | 209.59(38.88) | 0.50 | 193.93(37.26) | 193.59(38.98) | 0.95 |
| TG | 154.50(62.87) | 158.32(64.29) | 0.77 | 150.97(83.60) | 148.60(66.48) | 0.84 |
| HDL | 43.38(14.21) | 44.44(10.40) | 0.70 | 44.43(12.38) | 41.65(8.68) | 0.11 |
| LDL | 124.82(42.84) | 128.35(33.54) | 0.67 | 123.82(32.37) | 124.59(33.99) | 0.88 |
| FPG | 167.58(75.09) | 103.08(11.91) | **<0.001*** | 161.92(56.58) | 105.85(13.67) | **<0.001*** |
| PPPG | 230.61(91.94) | 133.69(38.06) | **<0.001*** | 213.35(92.29) | 121.89(23.32) | **<0.001*** |
| HbA1c | 9.03(2.49) | 5.99(1.48) | **<0.001*** | 8.10(1.59) | 5.81(0.58) | **<0.001*** |
| NT-proBNP | 497.93(129.10) | 453.25(52.33) | 0.056 | 301.49(87.68) | 278.69(96.44) | 0.11 |
| sST2 | 37.80(11.76) | 39.09(15.65) | 0.64 | 23.34(17.95) | 20.11(10.21) | 0.18 |
| Gal-3 | 24.39(10.51) | 25.96(11.54) | 0.49 | 16.69(8.07) | 18.50(8.39) | 0.16 |
| MiR-210-3p# | 1.31 (0.13-5.5) | 0.87(0.14-4.22) | 0.78 | 0.20 (0.09-0.71) | 0.30(0.13-0.93) | 0.47 |
| MAGGIC score | 23.97(5.86) | 20.94(5.58) | **0.014*** | 9.75(4.13) | 6.91(4.63) | **<0.001*** |

NYHA: New York Heart Association; TC: total cholesterol; TG: triglyceride; HDL: high-density lipoprotein, LDL: low-density lipoprotein; FPG: fasting plasma glucose; PPPG: post-prandial plasma glucose, HbA1c: Glycated hemoglobin; NT-proBNP: N-terminal- pro-brain natriuretic peptide, sST2: soluble suppression of tumorigenicity 2; Gal-3: Galectin-3; MiR: micro ribonucleic acid; MAGGIC: Meta-Analysis Global Group in Chronic Heart Failure.

Expressed as mean and SD; Student t-test was used; #: expressed as median and interquartile range; Mann-Whitney U test was used; @: Chi-square test was used;

**: p-value statistically high significant

*: p-value statistically significant.

**Table 6. The cut-off level of the biomarkers in patients with HF.**

| Variable | AUC | SE | p-value | 95%CI | Cut-off level | Sensitivity (%) | Specificity (%) |
|---|---|---|---|---|---|---|---|
| NT-proBNP | 0.65 | 0.03 | 0.001** | 0.58-0.71 | 355.5 | 64 | 46 |
| sST2 | 0.58 | 0.03 | 0.01* | 0.52-0.65 | 20.11 | 60 | 46 |
| Gal-3 | 0.51 | 0.03 | 0.60 | 0.44-0.58 | 17.58 | 55 | 57 |
| MiR-210-3p | 0.79 | 0.05 | <0.001** | 0.68-0.89 | 5.03 | 87 | 54 |
| MAGGIC score | 0.86 | 0.02 | 0.001** | 0.86-0.90 | 14 | 95 | 89 |

NT-proBNP: N-terminal- pro brain natriuretic peptide; sST2: soluble suppression of tumorigenicity 2; Gal-3: galectin-3; MiRNA: micro ribonucleic acid; MAGGIC: Meta-Analysis Global Group in Chronic Heart Failure.

**: p value statistically high significant

*: p value statistically significant.

predisposing the arteries to atherosclerosis. The atherosclerotic changes in the blood vessels could be aggravated by dyslipidemia. Studies done by Jarab et al, have shown that elevated LDL and decreased HDL is associated with reduced cardiac function. Inflammation associated with dyslipidemia plays a major role in the exacerbation of HF [26]. Yao et al, concluded that dyslipidemia induced by oxidative stress, inflammation, altered autophagy, microvascular density, and cardiomyocyte mitochondrial dysfunction lead to impaired cardiac structure and

**Table 7. The correlation among biomarkers in patients with HF.**

| Correlation Matrix | MAGGIC score | NT-proBNP | sST2 | Gal-3 | MiR-210-3p | TC | TG | HDL | LDL | HbA1c |
|---|---|---|---|---|---|---|---|---|---|---|
| **MAGGIC score** | – | | | | | | | | | |
| **NT-proBNP** | 0.676 <0.001** | — | | | | | | | | |
| **sST2** | 0.405 <0.001** | 0.416 <0.001** | _- | | | | | | | |
| **Gal-3** | 0.281 <0.001** | 0.231 <0.001** | 0.136 0.025* | – | | | | | | |
| **MiR-210-3p** | 0.136 0.167 | 0.067 0.497 | 0.087 0.377 | 0.160 0.102 | – | | | – | | |
| **TC** | 0.063 0.301 | 0.118 0.053 | 0.121 0.047* | 0.011 0.858 | 0.03 0.761 | – | | | | |
| **TG** | 0.016 0.799 | 0.003 0.963 | 0.012 0.848 | 0.078 0.203 | 0.173 0.077 | 0.337 <0.001** | – | | | |
| **HDL** | -0.081 0.183 | -0.04 0.515 | 0.098 0.108 | -0.061 0.318 | 0.015 0.876 | -0.501 <0.001** | -0.061 0.32 | – | | |
| **LDL** | 0.058 0.341 | 0.041 0.507 | 0.034 0.578 | 0.003 0.960 | 0.010 0.921 | 0.820 <0.001** | 0.339 <0.001** | -0.321 <0.001** | – | |
| **HbA1c** | 0.286 <0.001** | 0.268 <0.001** | 0.159 0.012* | 0.045 0.484 | 0.152 0.124 | 0.107 0.093 | 0.024 0.702 | -0.065 0.306 | 0.076 0.232 | – |

NT-proBNP: N-terminal- pro-brain natriuretic peptide, sST2: soluble suppression of tumorigenicity, MiR: micro ribonucleic acid.

**: P value statistically highly significant

*: P value statistically significant.

function. The measures adopted to decrease serum lipids are found to improve the ventricular functioning capacity [27].

HFpEF is found to have widespread proinflammation due to comorbid conditions such as DM, hypertension, obesity, and renal impairment. It has been found that combination of biochemical markers such as sST2, NT-proBNP, and Gal-3 could aid in identifying the pathophysiology of the disease, and thus implementation of appropriate management [28]. A study by Mitic et al. shows that sST2, growth differentiation factor (GDF)-15, syndecan-1 and Gal-3 are in very high levels in HFrEF compared to healthy controls, HFmrEF and HFpEF. These markers showed association with ECHO findings such as LVH and end-diastolic dysfunction [29]. According to Piek et al, ventricular myocyte stress stimulates the synthesis and release of natriuretic peptides (NPs) [30]. NPs can identify asymptomatic HF in individuals with poor LV function. BNP by regulating the renin-angiotensin-aldosterone system (RAAS) and the sympathetic nervous system, causes vasodilatation, natriuresis and diuresis [31,32].

In the present study, NT-proBNP levels were 482.74 ± 110.94 and 287.81 ± 93.45 pg/mL in groups 1 and 2 respectively (p = 0.001) (Table 2). This indicated that group 1 patients had more severe LV volume overload than those group 2 patients, probably due to poor LV function and decreased EF. The NP levels did not get altered among the groups when compared among genders (Table 3). When NT-proBNP levels were compared between the individuals with lesser- and greater than 50 years of age, individuals with age more than 50 years had higher levels which was statistically significant in HFrEF (p = 0.02) as well as in HFpEF (p = 0.02) compared to other markers (Table 4). Individuals with diabetes did not show significantly altered NT-proBNP levels. According to the study by Kang et al, although patients with HFpEF have low NT-proBNP levels, the prognosis is similar to that of HFrEF [33]. Thus, in non-acute conditions, and when ECHO is not accessible, measurement of NPs may help to establish an initial diagnosis [6]. But, NT-proBNP measurement has certain limitations, which

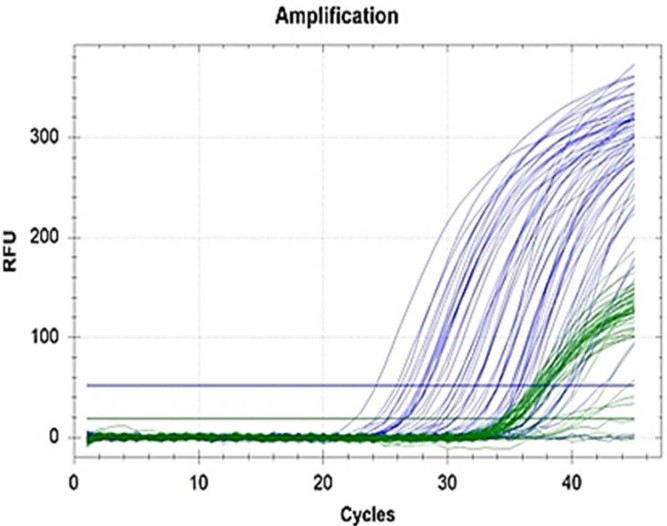

**Fig 2. MicroRNA amplification patterns among group 1 HF patients.**

preclude NPs from being used as a gold standard marker in all the situations. Cao et al, have concluded that there is no consensus in arriving at the age-adjusted NP cut-off levels [34]. Chronic renal failure, old age and smoking increase NT-proBNP in males, while T2DM and obesity decrease the levels in females [35].

The 2022 ACC Foundation/ AHA guidelines have recommended that in addition to markers of LV overload, biomarkers of inflammation and fibrosis may be used to refine the prognostic stratification of HF [36]. Biomarkers like sST2 and Gal-3 provide additional predictive information, particularly in individuals who are at risk for HF [37]. sST2 production

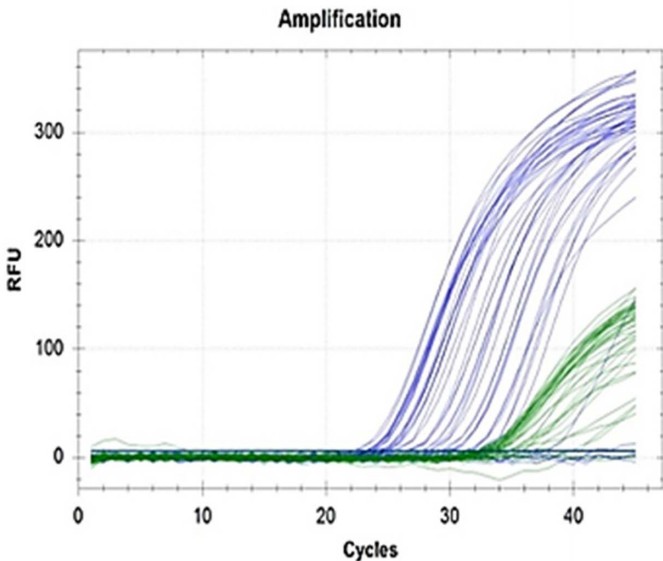

**Fig 3. MicroRNA amplification patterns among group 2 HF patients.**

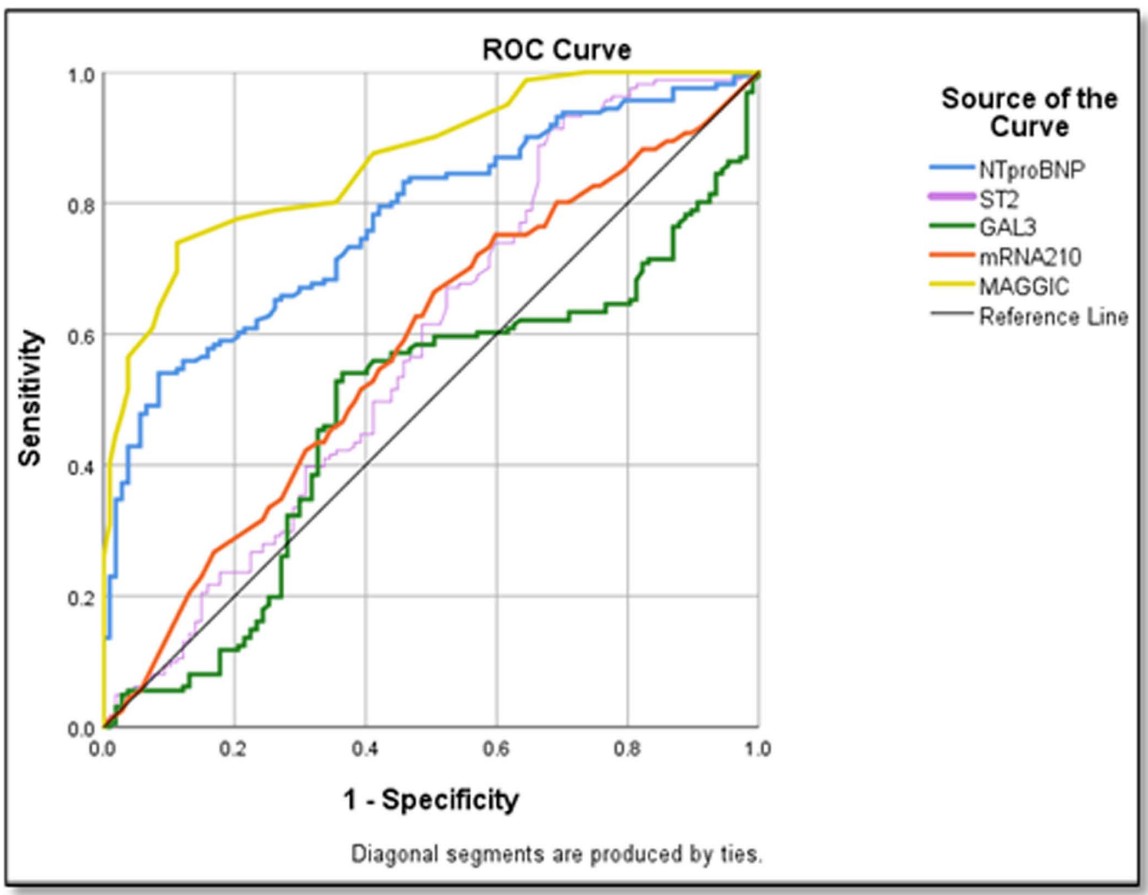

**Fig 4. The cut-off level of the biomarkers in patients with HF.**

is induced by mechanical stretch, inflammation, and fibrosis [38]. In the present study, sST2 levels were $38.23 \pm 13.14$ and $21.40 \pm 13.87$ ng/mL in groups 1 and 2 respectively (p = 0.001). This probably showed that sST2 indicated myocardial fibrosis, which could be higher in group 1 than in group 2 (Table 2). Elevated sST2 levels predict ventricular arrhythmia, LV diastolic dysfunction, increased RV systolic pressure, and hypokinesis in chronic HF [38]. sST2 levels rise several weeks before a cardiac event; thus, serving as a sensitive marker in the diagnosis of HF [11]. The lower biological variability of sST2, makes sST2 a promising marker for assessing the prognosis of HF patients [39]. sST2 is not affected by BMI, renal function, age, aetiology, or other traditional risk factors of HF [39]. In the present study, sST2 levels were not altered by gender (Table 3) and by age (Table 4). sST2 levels did not show statistically increased levels in diabetic patients compared to nondiabetics in both HFrEF and HFpEF (Table 5). According to Bai et al, sST2 and NT-proBNP levels are elevated in diabetic patients in HFpEF and they could serve as diagnostic markers [40].

According to Gehlken et al, Galectin-3 is highly stable and resistant to hemodynamic alterations, and is an early indicator of cardiac fibrosis, inflammation, ventricular remodeling, and renal impairment in HF patients; however, it has low specificity [41,42]. In the present study, serum Gal-3 levels were $24.93 \pm 10.84$ and $17.77 \pm 8.29$ ng/mL in groups 1 and 2 respectively (p = 0.001) (Table 2). Galectin-3 levels were not altered by gender (Table 3), age (Table 4) as well as the presence of diabetes mellitus (Table 5). Gal-3 levels are directly related to the

severity of diastolic dysfunction; thus, can detect early onset HFpEF and also predict prognosis in acute and chronic HF irrespective of EF [41,43].

Studies are being done to assess the diagnostic and prognostic potential of epigenetic markers such as micro-RNAs (miRNA), long non-coding RNAs, and exosome cargo, because of their potential role in the early phases of cardiac dysfunction. miRNAs play critical roles in heart development, and function, as well as response to ischemia/reperfusion injury. There is increasing evidence suggesting that that miRNAs may have key roles in the pathogenesis of HF through the regulation of genes involved in adverse ventricular remodelling [44]. The dysregulation of miRNAs, and their target genes, are related to signalling pathways of HF, including cardiac hypertrophy, oxidative stress, cardiac fibrosis, etc [45]. The circulating microRNAs are relatively stable, sensitive, specific, and easily detectable by non-invasive methodology, hence, circulating microRNAs may be ideal candidate biomarkers and drug targets for HF [45]. In response to hypoxia, a specific set of microRNAs, called hypoxamirs are upregulated [44].

In the present study, microRNA 210-3p showed increased expression of 11.89-fold in group 1 and 1.08-fold in group 2 compared to controls (p = 0.031) (Table 2, Figs 2 and 3). MiR-210-3p levels were not influenced by gender (Table 3), age (Table 4) and presence of diabetes mellitus (Table 5). According to Shen et al, the most down-regulated miRNA is miR-30c, and the most upregulated is miR-210-3p [45]. Through the HIF-1α pathway, miR-210-3p is independently associated with HF, cardiovascular mortality, and cardiac regeneration. MiR-210-3p is found to inhibit apoptosis of cardiomyocytes. HF patients who have increased expression levels of miR-210-3p show frequent HF decompensations and adverse ventricular remodeling [44]. MiRNAs can be used alone or in combination with other biomarkers of HF such as BNP [46].

The receiver operating characteristics (ROC) curve showed an area under the curve (AUC) for NT-proBNP of 0.65 (p = 0.001). The cut-off value of NT-proBNP was 355.5pg/mL with sensitivity and specificity of 64% and 46% respectively (Table 6, Fig 4). A normal electrocardiogram and/or plasma concentrations of BNP < 35 pg/mL and/or NT-proBNP < 125 pg/mL may be used in the diagnosis of HF [34]. In young healthy adults, BNP and NT-proBNP levels are < 25pg/mL and < 70pg/mL respectively [47]. If the BNP level is 100 - 500pg/mL, clinical findings are needed to support the diagnosis of HF. When BNP is > 500pg/mL, HF is present, and quick management for HF is needed [34]. The AUC for sST2 of 0.58 with a 95% CI of 0.52 to 0.65 (p = 0.01). The cut-off value of sST2 was 20.1ng/mL with sensitivity and specificity of 60% and 46% respectively. (Table 6, Fig 4) The study by Castiglione et al, shows a higher cutoff of 35 ng/mL for risk stratification in acute HF [48]. This is much higher than the cut-off of the present study probably due to variations in the ethnicity and the race of the study population. The present study showed almost the same cut-off as that of the PRIDE study. In the PRIDE study, sST2 concentration of more than 17.26 ng/mL has a stronger correlation with adverse prognosis in acute HF, and patients with increased sST2 levels show a high risk of death within a year [49].

In the present study, the ROC curve showed an AUC for Gal-3 of 0.51 with a 95% CI of 0.44 to 0.58; p = 0.60. The cut-off value of Gal-3 was 17.58 ng/mL with sensitivity and specificity of 55% and 57% respectively. (Table 6, Fig 4) According to Jiang et al, the cut-off value of Gal-3 to diagnose HF is 17.8 ng/mL. Gal-3 is higher in patients with HFpEF than that in HFrEF. Gal-3 levels are associated with ECHO findings, which further substantiates that Gal-3 is useful in diagnosing HFpEF [1]. Two studies show the same Gal-3 cut-off values are similar to the present study. In the DEAL study, after adjusting for confounders, the cut-off value of Gal-3 is 17.72ng/mL [50]. In the COACH study (Coordinating Study Evaluating Outcomes of Advising and Counseling in Heart Failure), Gal-3 values more than 17.8 ng/mL

are related to an increased risk of new hospitalization for HF [51]. On the contrary, in the HF-ACTION cohort, which includes ambulatory HF patients who underwent a structured exercise program, the median Gal-3 level is 14.0 ng/mL, which is lower than the present study [46].

Among the individual biomarkers, miRNA-210-3p showed the most promise as a novel biomarker, outperforming traditional markers with a higher AUC of 0.79, sensitivity of 87%, and specificity of 54% at a cut-off level of 5.03. (Table 6, Fig 4) MicroRNA210-3p had the highest sensitivity and AUC among the biomarkers analysed in the present study. MiR-210-3p could be an early diagnostic marker of HFpEF. The microRNA detection methods should ensure accuracy and precision [52]. MAGGIC score demonstrated the overall best performance with the highest AUC of 0.86 (95% CI: 0.86-0.90) (p = 0.001, a cut-off score of 14 with sensitivity of 95% and specificity of 89%, suggesting its reliability for risk stratification (Table 6, Fig 4).

The MAGGIC score showed a positive correlation with NT-proBNP, sST2, and galectin-3. However, miR-210-3p did not show any correlations with any of the variables, probably due to the small sample size. NT-proBNP, sST2, and MAGGIC score positively correlated with HbA1c (Table 7). Jiang et al, concluded that in HF, BNP is sensitive to volume overload, while Gal-3 is sensitive to fibrosis, hence a combination of these markers may be beneficial in diagnosis [1]. Patients with elevated levels of both NPs and Gal-3 have more than twice the risk of developing HF [46]. Cui et al, have shown that Gal-3 is superior to sST2 in distinguishing HFpEF from healthy individuals and other phenotypes of HF. However, sST2 provides incremental value to NT-proBNP in HF. Although AHA has recommended Gal-3 and sST2 to be used along with NPs for risk stratification in HF, only a few data are available concerning their prognostic value in HFpEF [53]. Since Gal-3 levels are directly related to LV end-diastolic pressure in HFpEF, elevated levels are associated with frequent hospitalizations and increased mortality [53]. Only Gal-3 maintains strong prognostic ability after correction for clinical factors and NPs [54]. Similar to NPs, Gal-3 is also limited by CKD with GFR < 60 ml/min/1·73m². Gal-3 is useful in predicting adverse events in HFpEF, whereas sST2 is useful in HFrEF. Serum sST2 measurements provide information on myocardial fibrosis and systolic dysfunction [53]. The combination of NT-proBNP and sST2 gives an insight into NYHA states. sST2 is relatively better than NT-proBNP in the detection of HFpEF in hypertensive patients [46]. Gal-3 has few advantages over the other two markers, in that it predicts HF risk in the general population as well as in the early fibrosis and ventricular remodelling in HF [13].

The pathogenesis of HF is highly complex, hence relying on a single biomarker will not be sufficient in diagnosing heart failure. Each biomarker has its inherent advantages and disadvantages. Thus, a multi-biomarker strategy, such that biomarkers could provide information on different pathophysiological pathways, may offer a promising strategy that goes beyond the limits of the current management of HF [55]. In community-based screening, ECHO may be supplemented with NT-proBNP; and sST2 and Gal-3 may be done in addition to stratifying the patients for effective management [56]. Also, it enhances the possibility of employing tailored therapies for individuals with heterogeneous presentations. ACC/AHA guidelines suggest that a combination of biomarkers is beneficial for risk stratification. It is better to have an integrated protocol of biomarkers with clinical variables and imaging characteristics [48]. However, based on the growing knowledge of the molecular properties of Gal-3 and its complex mechanism of action, as well as advancing knowledge of the pathophysiology of HF and the therapeutic approach, there is a space for tailoring medical procedures depending on the characteristics of the patient population and the time point of the developing pathology [43].

## Strengths

The study was carried out in India, where the resource settings are limited. India as such is highly diverse in economy, food, physical habits, and literacy rate across the various states. However, the study was carried out in a tertiary care hospital in a Metropolitan city. To a major extent, it takes into account the various factors. A multi-marker study in a relatively large sample size comprising both genders remains be major strength of the study.

## Limitations

The study did not include healthy individuals and hence, baseline levels of healthy individuals could not be established. Also, the influence of metabolic diseases- hypertension, diabetes mellitus, and obesity on HFpEF could not be established. A cohort of hypertensive and diabetes mellitus patients could establish the cause-and-effect relationship and a better profile of markers according to the stage of the study.

## Conclusion

MicroRNA-210-3p, NT-proBNP, sST2, and galectin-3 were markedly elevated in HF patients with EF less than 50% than in patients with more than 50%. The gold-standard marker- NT-proBNP showed a positive association with both sST2 and galectin-3. Also, there was a positive association of sST2 with galectin-3. MiR-210-3p fold changes were positively associated with galectin-3, indicating that both the markers indicate the cardiac fibrosis stage of HF. Since each marker has its specific advantages and disadvantages, a multi-marker approach could help in the early diagnosis as well as in stratifying HF patients so that appropriate target therapies can be implemented according to the pathogenic stage of the disease.

## Supporting information

**S1 File. Available from: https://doi.org/10.5061/dryad.qjq2bvqs4**

## Author contributions

**Conceptualization:** Jasmine Chandra Arul, Sudagar Singh Raja Beem, Mohanalakshmi Parthasarathy, Santhi Silambanan.

**Data curation:** Jasmine Chandra Arul, Sudagar Singh Raja Beem, Mohanalakshmi Parthasarathy, Mahesh Kumar Kuppusamy, Karthikeyan Rajamani, Santhi Silambanan.

**Formal analysis:** Sudagar Singh Raja Beem, Mohanalakshmi Parthasarathy, Mahesh Kumar Kuppusamy, Karthikeyan Rajamani, Santhi Silambanan.

**Funding acquisition:** Jasmine Chandra Arul, Santhi Silambanan.

**Investigation:** Jasmine Chandra Arul, Mohanalakshmi Parthasarathy, Karthikeyan Rajamani, Santhi Silambanan.

**Methodology:** Jasmine Chandra Arul, Sudagar Singh Raja Beem, Mahesh Kumar Kuppusamy, Karthikeyan Rajamani, Santhi Silambanan.

**Project administration:** Mahesh Kumar Kuppusamy, Santhi Silambanan.

**Resources:** Jasmine Chandra Arul, Santhi Silambanan.

**Supervision:** Mohanalakshmi Parthasarathy, Santhi Silambanan.

**Visualization:** Karthikeyan Rajamani.

**Writing – original draft:** Jasmine Chandra Arul, Sudagar Singh Raja Beem, Mohanalakshmi Parthasarathy, Mahesh Kumar Kuppusamy, Karthikeyan Rajamani, Santhi Silambanan.

**Writing – review & editing:** Sudagar Singh Raja Beem, Santhi Silambanan.

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
