## [Decision Letter · Decision Letter 0]

13 Aug 2024

PONE-D-24-15389Association of NT-proBNP, sST2, Galectin-3 and microRNA-210-3p in heart failure patients with preserved and reduced ejection fraction- cross-sectional study

PLOS ONE

Dear Dr. Silambanan,

Thank you for submitting your manuscript to PLOS ONE. After careful consideration, we feel that it has merit but does not fully meet PLOS ONE’s publication criteria as it currently stands. Therefore, we invite you to submit a revised version of the manuscript that addresses the points raised during the review process.

We look forward to receiving your revised manuscript.

Kind regards,

Tommaso Lomonaco, Ph.D

Academic Editor

PLOS ONE

3. Please remove your figures from within your manuscript file, leaving only the individual TIFF/EPS image files, uploaded separately. These will be automatically included in the reviewers’ PDF.

Reviewers' comments:

Reviewer's Responses to Questions

**Comments to the Author**

1. Is the manuscript technically sound, and do the data support the conclusions?

Reviewer #1: Yes

Reviewer #2: Partly

2. Has the statistical analysis been performed appropriately and rigorously? 

Reviewer #1: Yes

Reviewer #2: No

3. Have the authors made all data underlying the findings in their manuscript fully available?

Reviewer #1: Yes

Reviewer #2: Yes

4. Is the manuscript presented in an intelligible fashion and written in standard English?

Reviewer #1: Yes

Reviewer #2: Yes

5. Review Comments to the Author

Reviewer #1: The authors reported that NT-proBNP, sST2 and galectin-3 were significantly higher in HFrEF compared to that of

HFpEF. Also, microRNA-210-3p showed significantly increased expression in HFpEF compared to controls. The findings seem to be intresting, however, I would like to make some commnets.

1. Major concerns.

- The lack of the novelty is the concern that I would like to express around the article, while data for miR-210-3p decerve to be published.

- The authors did not validate the findings and the comparissons with widely used scores (MAGGIC, Barcelona-HF, etc.) were not performed

2. Minor concerns

- Structure of the paper is not logic. Section Results contains a paragraph for B-more Echo, while it could replaced to the subsection Methods. Please, check and correct.

- The authors did not report inclusion / exclusion criteria, flow chart for patients enrollement, comorbidity status, antropomorphic findings. Table 1 does not include the parameters mentioned above.

- Biomarkers evaluation and blood sampling. It remaines unclear now and in which ways the sampling were performed.

- Did the authors discovery other correlations between biomarkers and comorbidities, metabolic status, i.e. fasting glucose, HOMA-IR, lipid profile.

Reviewer #2: With the aim of investigating levels of biomarkers in HF and to compare between HFrEF and HFpEF, the authors performed a cross-sectional study enrolling 270 patients.

The authors are to commend for the great effort. However, there are some points in need of further clarification.

Specific points:

-Abstract:

Too much information about methods; please shorten it

Suddenly, the authors named in the result section controls. Please, add in the method.

-Introduction: too long. Please, go straight to the main focus of the paper.

-Methods: to include only Hypertensive patients with HF and then add patients with DM could be considered as a mistake.

-Results: the result section should include only results. Please, delete all the information that are not results.

Please, add also comparison with regard to sex (M vs F) and age.

-Discussion:

Too long and not focused on the main results. Pleas shorten it and make more focused on the main topic; in addition, add subparagraphs to improve readability.

To date, the role of biomarkers, and in particular the role of multimarkers, in HF can be considered in different stage (e.g., diagnosis, prognosis, management). Please comment using references (i.e., PMID 33673947 and 30832821)

Figures:

the figure are very low quality. Please increase.

Figure 1: please, miRNA and sST2 are too similar in color. Please change.

Consider to add in the text and in the discussion that one of the strengths of the present paper is that it has been performed in India (adding interesting additional data of patients whose ethnicity is classically not well investigated).

Additional comment:

If the authors used AI for the preparation of this manuscript, please declare.

6. PLOS authors have the option to publish the peer review history of their article (what does this mean? ). If published, this will include your full peer review and any attached files.

**Do you want your identity to be public for this peer review?** For information about this choice, including consent withdrawal, please see our Privacy Policy .

Reviewer #1: No

Reviewer #2: **Yes: ** Dr Andrea Salzano, MD, PhD

---

## [Author Response · Author response to Decision Letter 1]

28 Sep 2024

The authors are grateful for all the suggestions raised by the reviewers and editors. Herewith we are attaching the responses to the queries. We have taken utmost care and responsibility in addressing the queries. Once again thank you very much.

Responses to Journal requirements: 14-09-2024

Response: The manuscript has been modified according to the PLOS ONE’s requirements.

Response: The authors confirm that the data supporting the findings of this study are available within the article. Raw data were generated at the location of research, Sri Ramachandra Institute of Higher Education and Research, Chennai, India. No supplementary material is available with this research.

3. Please remove your figures from within your manuscript file, leaving only the individual TIFF/EPS image files, uploaded separately. These will be automatically included in the reviewers’ PDF.

Responses: Figures are removed from the manuscript file and uploaded separately as TIFF files.

Response to the reviewers comments:

1. Is the manuscript technically sound, and do the data support the conclusions?

Reviewer #1: Yes

Reviewer #2: Partly

Response: The manuscript has been edited with the responses to the reviewers, which we think could improve the content and presentation of the manuscript.

2. Has the statistical analysis been performed appropriately and rigorously?

Reviewer #1: Yes

Reviewer #2: No

Response: Correlation studies and ROC curve have been modified. The information is added in the methodology, results, discussion and conclusion.

3. Have the authors made all data underlying the findings in their manuscript fully available?

Reviewer #1: Yes

Reviewer #2: Yes

4. Is the manuscript presented in an intelligible fashion and written in standard English?

Reviewer #1: Yes

Reviewer #2: Yes

5. Review Comments to the Author

Reviewer #1: The authors reported that NT-proBNP, sST2 and galectin-3 were significantly higher in HFrEF compared to that of HFpEF. Also, microRNA-210-3p showed significantly increased expression in HFpEF compared to controls. The findings seem to be interesting, however, I would like to make some comments.

1. Major concerns.

- The lack of the novelty is the concern that I would like to express around the article, while data for miR-210-3p deserve to be published.

Response: The study was intended to be conducted since NT-proBNP levels do not correlate with the severity of disease especially in cases of HFpEF. Hence, we researched to find out whether sST2 and Gal-3 could be able to diagnose those cases which were not identified by NT-proBNP. Also, microRNAs could serve as very early biomarkers of the disease. Like we do profile of certain biomarkers in case of liver or renal diseases, a profile of biomarkers in HF will reduce the chances of morbidity and mortality. Of late women tend to develop hypertension almost similar to that of men. Studies show that women respond poorly to antihypertensives compared to men. Hence, the chances of increasing cases of HFpEF is high in future. Keeping all these in mind, we proceeded with the study.

- The authors did not validate the findings and the comparisons with widely used scores (MAGGIC, Barcelona-HF, etc.) were not performed

Methodology: The Meta‐Analysis Global Group in Chronic Heart Failure (MAGGIC) mortality risk score was calculated in all the study participants. The MAGGIC score consists of thirteen clinical predictor variables such as age, sex, body mass index (BMI), systolic blood pressure (SBP), ejection fraction (EF), serum creatinine level, participant is a current smoker, has diabetes mellitus, and chronic obstructive pulmonary disease, belongs to New York Heart Association (NYHA) class, heart failure (HF) diagnosed for at least eighteen months, use of beta-blockers and angiotensin‐converting enzyme (ACE) inhibitors.

2. Minor concerns

- Structure of the paper is not logic. Section Results contains a paragraph for B-mode Echo, while it could replaced to the subsection Methods. Please, check and correct.

Response: The following removed in results and added in methodology

The study participants were subjected to the transthoracic 2D doppler echocardiography using Phillips and GE Healthcare echocardiography with patients lying supine in the left lateral decubitus position. EF values in healthy individuals and in heart failure patients vary with ethnicity and type of ECHO instrument. According to the definition of European and United States guidelines, the normal EF is 52-72% in men and 54-74% in women [21]. According to Chengode, EF > 55% is considered to be normal. It can be 45%–54%, 30%–44% and < 30% in mild, moderate and severe LV dysfunction respectively [22].

Anthropometric characteristics such as body mass index (BMI) in Kg/m2 and waist to hip ratio (WHR) were measured as per the Asia-Pacific guidelines [23]. Systolic and diastolic blood pressures were measured using standard procedure and patients were identified as hypertensives based on the JNC7 blood pressure classification [24]. According to International Expert Committee, diabetes mellitus was diagnosed based on glycated hemoglobin (HbA1c) ≥6.5%, fasting plasma glucose ≥126 mg/dL and 2-hour plasma glucose or random plasma glucose ≥200 mg/dL [25].

LV ejection fraction (EF) remains the major parameter for phenotyping, diagnosis, prognosis and treatment decisions in HF. In the present study, EF of 50% was used as the cut-off to categorize the participants into two groups as group 1 – HF with EF ≤ 49% and group 2 -HF with EF � 50%. As per the 2016- ESC Guidelines, EF of 40- 49% is categorized as HFmrEF. HFmrEF is an intermediate clinical entity between HFrEF and HFpEF, but more similar to HFrEF with regard to the high prevalence of AMI in these patients [7]. Patients with HFmrEF are similar to HFrEF in being younger males, increased systolic blood pressure and reduced LV and atrial dilatation. Also, HFmrEF are found to benefit from HFrEF treatment strategies [7]. In the present study, based on the above literature data, HFmrEF has been included in HFrEF.

- The authors did not report inclusion / exclusion criteria, flow chart for patients enrolment, comorbidity status, anthropometric findings. Table 1 does not include the parameters mentioned above.

Response: Figure 1 shows the flow chart indicating the recruitment of participants into the study along with patient characteristics. Table 1 shows the demographic data of the study participants.

- Biomarkers evaluation and blood sampling. It remains unclear now and in which ways the sampling were performed.

Response: The heart failure patients who were attending cardiology department were recruited. Based on EF as shown by ECHO they were categorized as group I or 2. Subsequently blood samples were drawn for analysis of biomarkers and microRNA. Serum and plasma were separated and stored at -80�C until analysis.

- Did the authors discover other correlations between biomarkers and comorbidities, metabolic status, i.e. fasting glucose, HOMA-IR, lipid profile.

Response: The correlations of cardiac biomarkers with lipids, MAGGIC score and HbA1c are shown in table 6

Reviewer #2: With the aim of investigating levels of biomarkers in HF and to compare between HFrEF and HFpEF, the authors performed a cross-sectional study enrolling 270 patients. The authors are to be commended for the great effort. However, there are some points in need of further clarification.

Thank you very much for analyzing our manuscript critically. I am grateful for the valuable suggestions and giving us the opportunity to answer to the queries. Indeed, we are thankful and most obliged for all the constructive comments. Thank you

Specific points:

-Abstract:

Too much information about methods; please shorten it

Suddenly, the authors named in the result section controls. Please, add in the method.

Responses: Abstract -methodology has been modified. The information about controls has been added in the methodology

-Introduction: too long. Please, go straight to the main focus of the paper.

Response: Introduction was edited and excess information has been removed.

-Methods: to include only Hypertensive patients with HF and then add patients with DM could be considered as a mistake.

Response: Initially the study was decided to be conducted in hypertensive patients only, since hypertension has not been studied in Indian population. Due to the sudden onset of COVID-19 pandemic, patient visit to the hospital was limited. Hence, with the discussion and approval of research advisory committee members, it was decided to include hypertensives who also had diabetes mellitus. Ethics approval was sought again for the change in the criteria of recruitment of patients.

-Results: the result section should include only results. Please, delete all the information that are not results.

Results section has been modified and only results have been included. Information related to methodology have been added in methods section.

Please, add also comparison with regard to sex (M vs F) and age.

Response: Tables 3 and 4 have been added to compare the groups according to gender and age respectively.

-Discussion:

Too long and not focused on the main results. Pleas shorten it and make more focused on the main topic; in addition, add subparagraphs to improve readability.

Response: Discussion has been edited and shortened as per the suggestions.

To date, the role of biomarkers, and in particular the role of multimarkers, in HF can be considered in different stage (e.g., diagnosis, prognosis, management). Please comment using references (i.e., PMID 33673947 and 30832821)

Response: I had the opportunity to go through both the manuscripts. Both are really good, with good presentation and lot of information, pointing to almost the same findings which I have put forth in the manuscript. I have them in manuscript also.

Figures:

the figure are very low quality. Please increase.

Figure 1: please, miRNA and sST2 are too similar in color. Please change.

Response: The color has been changed. New figure with better clarity is attached

Consider to add in the text and in the discussion that one of the strengths of the present paper is that it has been performed in India (adding interesting additional data of patients whose ethnicity is classically not well investigated).

Response: Strength of the study has been added.

The study is carried out in India, where the resource settings are limited. India as such is highly diverse in economy, food, physical habits, literacy rate across the various states. But the study was carried out in a tertiary care hospital in a Metropolitan city. So to major extent it takes into account of the various factors. A multi-marker study in a relatively large sample size comprising of both genders remains to be major strength of the study.

Additional comment:

If the authors used AI for the preparation of this manuscript, please declare.

Response: The authors declare that AI for not used in preparation of this manuscript.

6. PLOS authors have the option to publish the peer review history of their article (what does this mean?). If published, this will include your full peer review and any attached files.

Do you want your identity to be public for this peer review? For information about this choice, including consent withdrawal, please see our Privacy Policy.

Reviewer #1: No

Reviewer #2: Yes: Dr Andrea Salzano, MD, PhD

---

## [Decision Letter · Decision Letter 1]

10 Dec 2024

PONE-D-24-15389R1Association of NT-proBNP, sST2, Galectin-3, and microRNA-210-3p in heart failure patients with preserved and reduced ejection fraction- cross-sectional studyPLOS ONE

Dear Dr. Silambanan,

Thank you for submitting your manuscript to PLOS ONE. After careful consideration, we feel that it has merit but does not fully meet PLOS ONE’s publication criteria as it currently stands. Therefore, we invite you to submit a revised version of the manuscript that addresses the points raised during the review process.

We look forward to receiving your revised manuscript.

Kind regards,

Tommaso Lomonaco, Ph.D

Academic Editor

PLOS ONE

Reviewers' comments:

Reviewer's Responses to Questions

**Comments to the Author**

1. If the authors have adequately addressed your comments raised in a previous round of review and you feel that this manuscript is now acceptable for publication, you may indicate that here to bypass the “Comments to the Author” section, enter your conflict of interest statement in the “Confidential to Editor” section, and submit your "Accept" recommendation.

Reviewer #1: All comments have been addressed

Reviewer #3: (No Response)

2. Is the manuscript technically sound, and do the data support the conclusions?

Reviewer #1: Yes

Reviewer #3: Partly

3. Has the statistical analysis been performed appropriately and rigorously? 

Reviewer #1: Yes

Reviewer #3: No

4. Have the authors made all data underlying the findings in their manuscript fully available?

Reviewer #1: (No Response)

Reviewer #3: Yes

5. Is the manuscript presented in an intelligible fashion and written in standard English?

Reviewer #1: Yes

Reviewer #3: Yes

6. Review Comments to the Author

Reviewer #1: The authors re-submitted the critically revised version of the manuscript with clear responds to reviewer. I am satisfied with the revision and have no serious concerns about the paper in its revised version.

Reviewer #3: This word document with the instruction for the authors has been uploaded, and also I have uploaded the PDF with the examples where some of the corrections should be made.

7. PLOS authors have the option to publish the peer review history of their article (what does this mean? ). If published, this will include your full peer review and any attached files.

**Do you want your identity to be public for this peer review?** For information about this choice, including consent withdrawal, please see our Privacy Policy .

Reviewer #1: No

Reviewer #3: **Yes: ** Dijana Stojanovic, MD, PhD

---

## [Author Response · Author response to Decision Letter 2]

14 Jan 2025

Response to the reviewers’ comments: 28-12-2024

Reviewer 3:

Comments to the authors and responses from the authors

Thank you very much for the suggestions for improvement of the quality of the manuscript.

Query: Page 17: Already said

Response: Extra information/repetitions have been removed.

Query: Page 17: There are 2022 Guidelines, which includes new definition of HFmrEF.

Still there is no need to explain HFmrEF in the paragraph regarding patients and methods

Response: Updated the ref 7. Extra information/ reference has been removed

Ref 7: McDonagh TA, Metra M, Adamo M, Gardner RS, Baumbach A, Böhm M, et al. 2023 Focused Update of the 2021 ESC Guidelines for the diagnosis and treatment of acute and chronic heart failure: Developed by the task force for the diagnosis and treatment of acute and chronic heart failure of the European Society of Cardiology (ESC) With the special contribution of the Heart Failure Association (HFA) of the ESC. European Heart Journal. 2023 Oct 1;44(37):3627-39.

Query: Page 19: Table 1 is not clear enough, clarify as much as possible

Response: Table 1 has been edited and it is easily understandable now

Query: page 20: Avoid too many information, for easier reading and clarity

Response: Majority of patients were males and in group 1 they were in the older age of 61-75 years, whereas in group 2, majority of them were middle-aged persons. Most of them were either overweight or obese in both the groups (Table 1).

Query: page 20: Avoid to mention data which are already in the Table, but lack statistical significance

Response: The results have been edited and made simple and easily readable.

Query: Provide the p value (even it is mentioned in the Table)

Response: According to NYHA class, participants in group 1 were in NYHA classes III and IV, whereas in group 2, the participants were in classes I and II (p<0.001) (Table 1).

Query: Table 2. The levels of biomarkers...Please avoid the verb "show" (already explained in the text, you just have to name the Table)

Response: Changed to - The levels of biomarkers among the participants

Query: page 21: The same as above

Response: Table 3: The levels of biomarkers among the participants based on gender

Query: Page 22: Please, explain all abbreviations. The Table is pretty confused, should be clarified for better understanding

Response: Tables 1,3,4 and 5 have been edited. Abbreviations have been explained.

Query: Page 23: Each time when there is a statistically significant result please insert a p value.

Response: Each time when there is a statistically significant result, p value has been added

Query: Page 24: I believe that this should be marked as statistically significant.

Response: It has been marked as statistically significant

Query: page 24: Provide the p value if statistically significant

Response: p-value has been added

Query: Each time when you have statistically significant result, please provide a p value, to emphasize the relevance of your findings.

Response: p-value has been added

Query: page 25: Clarify better: HF1 and HF2 for instance

Response: Group 1: HF with EF≤less than 49%

Group 2 HF with EF more than 50%

Query: This should be elaborated in the discussion part. In the paragraph "Results" only results should be mentioned, without additional analysis.

Response: It has been deleted in results and added in discussion

Query: page 26: Please, bold statistically significant values. or insert * through entire Table

Response: * has been added in the table for significant p values

Query: Avoid description in the Results.

Response: Descriptions have been removed in results and added in discussion

Query: page 27: This should be included in the Table 1, as clinical features, or else

Response: presenting clinical features added in table 1

Query: Already mentioned, avoid repetition

Response: Repetitions have been removed

Query: This cant be claimed this without statistical analysis!

Response: In this study, most of the participants in both the groups were either overweight or obese. This could be the cause or consequence of HF.

Query: Irrelevant for discussion

Response: Removed- According to Cao et al, BNP levels help clinicians to distinguish whether the dyspnea is of cardiac or noncardiac origin [40].

Query: Please include the latest ESC GUIDELINES or 2022 AHA/ACC/HFSA Guideline

Response: Recent ref has been added as Ref 7

Query Page 30: ST2 has been excessively elaborated, it is not novel biomarker.

Response: Excess elaboration on sST2 has been removed.

Query: This assumption cant be made solely based on the difference between the ST2 concentration

Response: There is clear decrease in EF less than 49. The values are associated with NT-proBNP also.

Query: Abstract: It is filled with unnecessary information; make it shorter.

Response: We have edited the unnecessary information and made the abstract shorter.

Background:

Heart failure (HF) is a growing health problem, and around two percent of the general population is affected. Accurate diagnostic markers that have the potential for early diagnosis of HF are lacking. This study aimed to compare the expression levels of microRNA-210-3p with biomarkers NT-proBNP, sST2, and galectin-3 in heart failure patients with preserved and reduced ejection fractions.

Materials and Methods

The cross-sectional study was conducted on 270 hypertensive heart failure patients in the age group of 30 to 75 years of both genders. The participants with evidence of HF were recruited from the Department of Cardiology in a tertiary care hospital in Chennai, India. QRT-PCR analyzed MicroRNA-210-3p in a stratified sample of 80 HF patients and 20 healthy individuals. ELISA analyzed biomarkers. Institutional ethics committee approval and written informed consent were obtained. Statistical analysis was performed using R software (4.2.1). Appropriate statistical tools were used based on the type of data distribution. p-value ≤ 0.05 was considered to be statistically significant.

Results

All the biomarkers, including microRNA-210-3p, were significantly higher in HFrEF than in HFpEF. The MAGGIC score showed a positive correlation with all the biomarkers. The cut-off of miRNA-210-3p was 5.03.

Conclusion

All the biomarkers were significantly elevated in HFrEF compared to HFpEF. However, microRNA-210-3p could be an early marker in diagnosing heart failure. Employing a multi-marker approach could help in the early diagnosis and stratification of HF patients.

Query: Introduction: It does not focus to the main subject of the paper (this introduction includes already known information, focus mostly on microRNAs)

Response:

Heart failure (HF) is a complex multisystem disorder due to structural and functional alterations in the heart [1]. It affects 64.34 million people, leading to 9.91 million years lost due to disability (YLDs). In the US, HF prevalence has been projected to increase from 2.42% in 2012 to 2.97% in 2030 [2]. The prevalence of HF in Asia is 1.3 - 6.7%, which is higher than that of Western countries [3]. Among the Asian countries, the prevalences are 1.3%, 1.0%, 6.7%, and 4.5% in China, Japan, Malaysia, and Singapore respectively [4]. In India, the number of new HF cases may increase from 0.118 - 0.708 million in 2000 to 0.214 -1.3 million in 2025 [5].

HF patients are grouped into three phenotypes based on the ejection fraction (EF) of the left ventricle (LV), as measured by echocardiography (ECHO). HF is categorized as HF with reduced EF (HFrEF) with EF≤40%, HF with preserved EF (HFpEF) with EF�50%, and HF with mid-range/ mildly reduced EF (EFmrEF) with EF 41-49% [6]. HFrEF is preceded by acute myocardial infarction (AMI), valvular, and other cardiac diseases [7]. Hypertension is the most important cause of HFpEF, with a prevalence of 60 - 89% among HF patients [7]. HFpEF is prevalent among the older population, especially women. With the advent of new diagnostic and treatment modalities, the prevalence of HFrEF is declining, while that of HFpEF shows an increasing trend [8].

The diagnosis of HF is based on the Framingham Diagnostic Criteria for HF [9]. Diagnosis and management of HFrEF have been studied in detail and have been implemented effectively in practice [8]. However, HFpEF and HFmrEF are poorly investigated and managed, particularly in developing countries [10]. The European Society of Cardiology (ESC) guidelines (2016) recommend that the patients suspected to have acute HF should have their plasma NPs tested. However, the chances of missing the early-stage HF could be high, especially in situations where EF is more than 50% as in HFpEF [7]. American Heart Association (AHA)/ American College of Cardiology (ACC) guidelines recommend the measurement of soluble suppression of tumorigenicity 2 (sST2) for risk stratification in patients with acute HF [13]. Galectin-3 (Gal-3) is involved in inflammation, repair, and fibrosis in HF [14].

Despite the introduction of various markers, there is lot of chance of missing HF due to heterogeneity in the presentation of HF. Recent studies have shown that circulating microRNA (miRNA) is playing a crucial role in the pathogenesis and progression of several cardiac diseases, including HF. They have a great potential to be diagnostic as well as prognostic markers. According to Rincón et al. there is an association between the microRNAs such as miR‐210‐3p, miR‐221‐3p, and miR‐23a‐3p and cardiac morbidity and mortality [63]. MicroRNA-210-3p is upregulated in tissue hypoxic conditions; thus, regulating cell differentiation, proliferation, migration, apoptosis, mitochondrial metabolism, and angiogenesis [15]. Studies on the utility of miRNAs in the diagnosis and prognosis of HFpEF are limited. This study was aimed to determine the association of microRNA-210-3p, and biomarkers such as NT-proBNP, sST2, and Gal-3 in heart failure patients with preserved and reduced ejection fractions.

Query: Methods: The clinical group should be regarded as a whole, with the two subgroups with and without DM

Response:

The mean age of the participants with diabetes was higher than nondiabetics in both HFrEF (p=0.039) and HFpEF (p=0.040). Number of individuals with diabetes was higher than nondiabetics and most of them were males (p=0.003) and (p=0.004) in HFrEF and HFpEF respectively. The glycemic variables such as fasting plasma glucose (p<0.001), postprandial plasma glucose (p<0.001), and glycated hemoglobin (p<0.001) in both the groups. All the cardiac biomarkers including MiR-210-3p did not show statistically significant difference between diabetics and non-diabetics in HFrEF and HFpEF. The MAGGIC showed higher values in diabetics compared to nondiabetics in HFrEF (p=0.014) and HFpEF (p<0.001) (Table 5).

Query: Results: This section is very slopy and not very clear, please stick only to the data, not its elaboration or clarification. Each time that you have statistically significant results, emphasize it with “p” value, where there is no statistical significance do not repeat it again, it is already stated in the Tables.

Response:

Results section has been modified. Redundant information has been edited

Query: Discussion: This part should be rewritten completely, since it contains too much data and information and it is not clear for the readers. Also, the authors should include the newest guidelines, whether ESC or AHA, since mentioning 2013, or 2017 guideline is not of relevance. Still there is much said about markers of fibrosis and inflammation, including BNP. Therefore, the authors should emphasize microRNA and to mention other biomarkers just to make the comparison with microRNA, which should be the most important and the key point of the study (see the references PMID: 37734333 and PMID: 32295985)

Response: The updated guidelines have been added. Discussion has been rewritten with removal of unwanted information.

Query: Figures and Tables: Please, make them as clear as possible for better reading and understanding

Response:

Figures and tables have been edited for clear understanding

---

## [Decision Letter · Decision Letter 2]

18 Feb 2025

Association of microRNA-210-3p with NT-proBNP, sST2, and Galectin-3 in heart failure patients with preserved and reduced ejection fraction- a cross-sectional study

PONE-D-24-15389R2

Dear Dr. Santhi Silambanan,

We’re pleased to inform you that your manuscript has been judged scientifically suitable for publication and will be formally accepted for publication once it meets all outstanding technical requirements.

Kind regards,

Tommaso Lomonaco, Ph.D

Academic Editor

PLOS ONE

Reviewers' comments:

Reviewer's Responses to Questions

**Comments to the Author**

1. If the authors have adequately addressed your comments raised in a previous round of review and you feel that this manuscript is now acceptable for publication, you may indicate that here to bypass the “Comments to the Author” section, enter your conflict of interest statement in the “Confidential to Editor” section, and submit your "Accept" recommendation.

Reviewer #1: All comments have been addressed

2. Is the manuscript technically sound, and do the data support the conclusions?

Reviewer #1: Yes

3. Has the statistical analysis been performed appropriately and rigorously? 

Reviewer #1: Yes

4. Have the authors made all data underlying the findings in their manuscript fully available?

Reviewer #1: Yes

5. Is the manuscript presented in an intelligible fashion and written in standard English?

Reviewer #1: Yes

6. Review Comments to the Author

Reviewer #1: I am conpletely satisfied with the revision provided by the authors. I have no serious concerns about the paper in its revised version.

7. PLOS authors have the option to publish the peer review history of their article (what does this mean? ). If published, this will include your full peer review and any attached files.

**Do you want your identity to be public for this peer review?** For information about this choice, including consent withdrawal, please see our Privacy Policy .

Reviewer #1: No

---

## [Editor Report · Acceptance letter]

PONE-D-24-15389R2

PLOS ONE

Dear Dr. Silambanan,

I'm pleased to inform you that your manuscript has been deemed suitable for publication in PLOS ONE. Congratulations! Your manuscript is now being handed over to our production team.

Kind regards,

on behalf of

Prof. Tommaso Lomonaco

Academic Editor

PLOS ONE